# Mg^2+^ and Cu^2+^ Charging Agents Improving Electrophoretic Deposition Efficiency and Coating Adhesion of Nano-TbF_3_ on Sintered Nd-Fe-B Magnets

**DOI:** 10.3390/ma16072682

**Published:** 2023-03-28

**Authors:** Weitao Zhang, Yang Luo, Xiaojun Sun, Ze Zhang, Haijun Peng, Yuanfei Yang, Wenlong Yan, Zilong Wang, Dunbo Yu

**Affiliations:** 1National Engineering Research Center for Rare Earth, Grirem Advanced Materials Co., Ltd., Beijing 100088, China; 2Grirem Hi-Tech Co., Ltd., Beijing 100088, China; 3China General Research Institute for Nonferrous Metals, Beijing 100088, China

**Keywords:** sintered Nd-Fe-B magnet, grain boundary diffusion, electrophoresis deposition, charging agents

## Abstract

In order to prepare nano-TbF3 coating with high quality on the surface of Nd-Fe-B magnets by electrophoretic deposition (EPD) more efficiently, Mg^2+^ and Cu^2+^ charging agents are introduced into the electrophoretic suspension and the influence on the electrophoretic deposition is systematically investigated. The results show that the addition of Mg^2+^ and Cu^2+^ charging agents can improve the electrophoretic deposition efficiency and coating adhesion of nano-TbF3 powders on sintered Nd-Fe-B magnets. The EPD efficiency increases by 116% with a relative content of Mg^2+^ as 3%, while it increases by 109% with a relative content of Cu^2+^ as 5%. Combining the Hamaker equation and diffusion electric double layer theory, the addition of Mg^2+^ and Cu^2+^ can change the zeta potential of charged particles, resulting in the improvement of EPD efficiency. The relative content of Mg^2+^ below 3% and Cu^2+^ below 5% can increase the thickness of the diffusion electric double layer, the excessive addition of a charging agent will compress the diffusion electric double layer, and thicker diffusion layer represents higher zeta potential. Furthermore, the addition of Mg^2+^ and Cu^2+^ charging agents greatly improves the coating adhesion, and the critical load for the cracking of the coating increases to 146.4 mN and 40.2 mN from 17.9 mN, respectively.

## 1. Introduction

With their excellent magnetic performance, Nd-Fe-B magnets have been widely used in consumer electronics, automobile and wind power generation, etc. [1,2]. However, the low Curie temperature of a sintered Nd-Fe-B magnet limits its application in the field of high temperature. To obtain high-coercive Nd-Fe-B magnets, the heavy rare earth element (HRE) of Tb or Dy is added into the magnets because the (Nd, HRE)_2_Fe_14_B phase has a higher anisotropy field H_A_ than that of the Nd_2_Fe_14_B phase [3,4]. Grain boundary diffusion (GBD) is a commonly used technique to infiltrate heavy rare earth along the grain boundary from the surface into the interior, which only consumes a very small amount of HREs [5,6].

To prepare the diffusion source on the magnet surface, we can use different coating methods such as dipping [7], magnetron sputtering [8], Plasma Spraying [9], vapor deposition [10] and electrophoretic deposition (EPD) [11]. Charged particles of a suspension are deposited electrokinetically by an electric field placed between two oppositely charged electrodes; this process is called EPD. EPD has the advantage of having a lower cost than magnetron sputtering. Compared with coating and dipping, EPD offers many advantages, such as good shape freedom, uniform film, controllable thickness of film and relatively good adhesion [12]. In EPD, the suspension properties affect the EPD process and the quality of the final deposited coatings. The water-based suspensions cause problems in electrophoretic forming, such as gas generated by water electrolysis and joule heating of the suspension, which reduces the stability of the suspension [13]. Therefore, at present, HRE compounds diffusion sources such as TbF_3_, TbH_3_, DyF_3_ and DyH_3_ are mainly electrophoretically deposited in organic suspension [14,15,16,17]. Actually, organic liquids usually have a low dielectric constant, which limits the charges attached to the particles, and leads to the formation of electronically charged particle flocculation [13]. 

The dissociation or ionization of surface groups on particles and the adsorption of surfactants are the most important mechanisms to obtain stable nonaqueous EPD suspension and a good EPD process [18]. Charging agents including acids (H^+^), bases (OH^−^), adsorbed metal ions or adsorbed poly-electrolytes are used to obtain particle surfaces’ electrosteric stabilization for an effective EPD process [19]. Das et al. [20] prepared the stable suspension of yttria stabilized zirconia (YSZ) nanoparticles by using phosphate ester (PE) as a charging agent and revealed that the most stable suspension of YSZ nano-powder can be obtained when the PE concentration is 0.01 g/100 mL, because of the high zeta potential. Zarbov et al. [21] added Polyethyleneimine (PEI) into the BaTiO_3_ EPD suspension as a charging agent and found that PEI maintains its very strong cationic charge by protonation of the amine groups from the surrounding medium. Guan et al. [22] prepared Dy_2_O_3_ film on the magnet surface by EPD and used polyethylene imine (PEI) to improve the coating adhesion and EPD efficiency in an isopropanol suspension. Wang et al. [23] added 5 wt.% MgCl_2_ into TbF_3_ EPD suspension to improve the EPD efficiency and the coating adhesion. Cu^2+^ has also acted as a charging agent in the process of co-deposition to form copper–graphene composite films [24]. These studies focused on improving the EPD process or magnet property by using charging agents, but the effect of different relative content of charging agents on deposition has not been studied. Moreover, it is also important to explain the effect of metal ions on EPD efficiency from the perspective of deposition kinetics. In order to improve the efficiency of EPD and the coating adhesion of nano-TbF3 powders so as to prepare for the grain boundary diffusion process of the sintered magnets, the effect of different relative contents of Mg^2+^ and Cu^2+^ charging agents on EPD efficiency and coating adhesion were systematically studied. In addition, the mechanism of action of Mg^2+^ and Cu^2+^ charging agents was analyzed with the theory of EPD and diffusion double layer from the perspective of deposition kinetics.

## 2. Experimental

A commercial sintered Nd-Fe-B magnet provided by GRIREM Co., Ltd. was selected as the initial magnet. The magnet was wire cut into cuboids with sizes of 8 × 8× 7 mm (c-axis). Then, the magnet was polished with 600 mesh, 1000 mesh and 2000 mesh sandpaper until the surface of the sample was flat and smooth. Subsequently, the surface of the polished magnet sample was cleaned with conduct alkali washing, acid washing and anhydrous ethanol ultrasonic cleaning and dried for later use. Nano TbF_3_ (900 nm) powder provided by GRIREM Advanced Materials Co., Ltd. (Beijing, China), was used as the diffusion source. TbF_3_ powder and anhydrous ethanol were mixed into 8 g/L suspension, MgCl_2_ or CuCl_2_ with a mass of 1–10% relative to TbF_3_ was added and mechanical stirring was performed until MgCl_2_ or CuCl_2_ was completely dissolved; then, the ultrasonic was performed for 2 min until the suspension was uniform. Copper plate as anode and sample as cathode which were soaked in suspension, and TbF_3_ coating was obtained by EPD at 60 V voltage for 1–5 min. The deposited samples were then diffused under vacuum at 900 °C for 7 h, followed by tempering at 500 °C for 2 h.

The magnetic properties of the samples were measured by a high temperature permanent magnet measuring instrument (NIM-500C, Beijing, China) and the microstructure of the coating was characterized by a scanning electron microscope (SEM, Tescan Vega II, Brno, Czech Republic). The coating elements were analyzed by ICP. The phases of the coating powders stripped from the surface of magnet were measured by XRD. The viscosity of the suspension was measured by a rotary rheometer (Anton paar MCR302, Graz, Austria). The *ζ* potential of the suspension was tested by a Zeta potential tester (Brookhaven 90plus, New York, NY, USA). The acidity and alkalinity of the suspension were detected by precision pH test paper. Coating adhesion was tested by a micro-nano mechanical testing system (STEP500-NHT3-MCT3, Anton paar, Graz, Austria).

## 3. Results and Discussion

### 3.1. EPD Efficiency

Figure 1 shows the EPD rate of nano-TbF_3_ with different relative contents of Mg^2+^ (a) and Cu^2+^ (b). It was found that with the increase in the relative content of Mg^2+^, the EPD efficiency increased first and then decreased, reaching the highest value when the relative content of Mg^2+^ was 3%, with the efficiency improving 116% from 3.1 mg/(cm^2^/min) to 6.7 mg/(cm^2^/min). The voltage was used for ion migration when adding too much Mg^2+^, which resulted in EPD basically unable to occur. For Cu^2+^, the EPD efficiency also first increased and then decreased with the increase in the relative content of Cu^2+^. Unlike Mg^2+^, the EPD efficiency reached the highest when the relative content of Cu^2+^ was 5%, with the efficiency improving 109% from 3.1 mg/(cm^2^/min) to 6.47 mg/(cm^2^/min). EPD can continue to occur if Cu^2+^ is continuously increased, but the efficiency of EPD will decline rapidly. The reasons will be added later.

The EPD behavior with different time was studied by adding the best relative content of Mg^2+^ and Cu^2+^. Figure 2 shows the EPD TbF_3_ amount (a) and EPD rate (b) of nano-TbF_3_ at different times. It can be seen in Figure 2a that the amounts of EPD all increased linearly with time when EPD was measured from the beginning to three minutes, which is consistent with the fact that the amount of EPD TbF_3_ amount is proportional to the time in the EPD theory. It is also found in Figure 2b that the EPD rate reduced when the EPD time reached 4–5 min because the deposited EPD coating increases the resistance. However, the EPD rate with Cu^2+^ reduced more than that of the others, which indicates that some changes occur to reduce the EPD efficiency when adding Cu^2+^ to the EPD suspension. Relatedly, adding Mg^2+^ will not affect the EPD process.

Figure 3 shows the cross-sectional SEM morphologies of the EPD 90s coated magnet. It can be seen in Figure 3a that the thickness of the coating was about 40 μm. However, Figure 3b,c shows that the thickness increased to about 80 μm with 3% Mg^2+^ added and increased to 60 μm with 5% Cu^2+^ added. This also proves that the addition of Mg^2+^ and Cu^2+^ improves the EPD efficiency. 

In order to further study the effect of Mg^2+^ and Cu^2+^ on the EPD efficiency from the perspective of kinetics, we first introduce the EPD Hamaker equation [25]:(1)$$m=\frac{\varepsilon_{0}\varepsilon_{r}}{1.5\eta }C\zeta \frac{E}{L}t$$

In the equation, *m* is the amount of EPD, ε_0_ is the dielectric constant of vacuum, ε_r_ is the dielectric constant of solution, *η* is the viscosity of EPD suspension, *C* is the particle concentration in EPD suspension, *ζ* is the zeta potential of charged particles, *E* is the applied electric field strength, *L* is the electrode spacing and *t* is the EPD time. In this paper, both the applied electric field’s strength and the distance between the electrodes were constant. EPD efficiency was calculated by dividing the EPD quantity by the EPD time. Therefore, the EPD efficiency at the same EPD time is inversely proportional to the *η* and *ζ*. Firstly, the effect of Mg^2+^ and Cu^2+^ on the viscosity of the EPD suspension was tested in order to study the kinetic reason of improving the EPD efficiency. The rotating rheometer tested the viscosity of suspension by shearing generated by a rotating motion, so we tested the viscosity with a shear rate of 0–1000 to enhance the accuracy of the test. Figure 4 shows the viscosity of the EPD suspension with different Mg^2+^ and Cu^2+^ relative contents. It can be found that the viscosity of the EPD suspension increased with the increase in the shear rate (*γ*) because of the viscous resistance of the particles in the suspension, but the viscosity curves of different Mg^2+^ and Cu^2+^ relative contents were basically coincident. Since the average viscosity of the various concentrations was essentially consistent across the illustrations, it is clear that the addition of Mg^2+^ and Cu^2+^ had little impact on the EPD suspension’s viscosity.

Subsequently, the effect of Mg^2+^ and Cu^2+^ addition on the zeta potential of EPD suspension was investigated. Figure 5 shows the dependence of the zeta potential of EPD suspension on the relative contents of Mg^2+^ (a) and Cu^2+^ (b). It can be seen that the zeta potential initially increased and then decreased with the change of Mg^2+^ and Cu^2+^ relative content, which is consistent with the change trend of EPD efficiency in Figure 1. Similarly, zeta potential also reached the peak value when adding about 3% Mg^2+^ and 5% Cu^2+^ compared with Figure 1. According to the EPD Hamaker equation, the EPD efficiency is proportional to the *ζ* potential. The effect of Mg^2+^ and Cu^2+^ on *ζ* potential was highly consistent with that on EPD efficiency. Therefore, we can confirm that the adding of Mg^2+^ and Cu^2+^ affected the efficiency of EPD by changing the zeta potential of the EPD suspension.

In order to explain the lower rate of later EPD with the addition of Cu^2+^, or the higher zeta potential with a higher relative content of Cu^2+^ (such as 10%) than without Cu^2+^ but with a lower EPD rate than without Cu^2+^, we analyzed the coating elements. Table 1 shows the ICP element analysis results of coating; it was found that the coating deposited by adding Cu^2+^ contained 0.71 wt.% Cu, while adding Mg^2+^ contained none. Figure 6 shows the XRD results of EPD coating with Cu^2+^ charging agents; it can be seen that the coating had a peak of copper and no peak of CuCl_2_, which indicates that Cu^2+^ in EPD suspension will be reduced to Cu under the electric field. In addition, Mg^2+^ can hardly be reduced to elemental because the discharge order is after H^+^, while Cu^2+^ is in front of H^+^.
Anode: Cu − 2e^−^ = Cu^2+^
Cathode: Cu^2+^ + 2e^×^ = Cu

This process reduces the EPD rate because of consuming Cu^2+^ and reducing the content of positive charge in the EPD suspension. Additionally, the process of generating Cu is not conducive to the deposition of TbF_3_ on the surface of the magnet. This is also the reason for the uneven coating surface in Figure 3c.

According to the diffusion double layer theory [26], Mg^2+^ and Cu^2+^ affect the zeta potential of the EPD suspension by adsorption. Figure 7 shows a schematic of the diffusion electric double layer region of a TbF_3_ particle. It can be seen in Figure 7a that when no charging agents are added, the particle surface mainly absorbs the positive and negative ions produced by dissociation in suspension. It can be seen in Figure 7b that when the relative content of Mg^2+^ and Cu^2+^ was low, the surface charge of the ion was gradually saturated. The diffusion double layer gradually became thicker with the increase in Mg^2+^ and Cu^2+^ on the surface, and the zeta potential was also higher, so the EPD efficiency gradually increased. When EPD efficiency reached the highest value, the surface charge of the particles was saturated; as Figure 7c shows, the diffusion electric double layer was compressed by adding charging agents, so the ζ potential decreased with the increase in Cl^−^ in the suspension.

### 3.2. Coating Adhesion

Good coating adhesion can ensure the GBD result after EPD. To test it, we applied a linear normal load of transverse movement to the coating, and determined the coating adhesion according to the load when the coating cracked. Figure 8 shows the scratch test results of EPD coating. It was found that the critical load for cracking of the coating without a charging agent was 17.9 mN, but it increased to 146.4 mN and 40.2 mN when Mg^2+^ and Cu^2+^ were added, respectively, which indicates that the addition of Mg^2+^ and Cu^2+^ charging agents greatly improved the coating adhesion. Figure 9 shows the surface SEM morphologies of the EPD 90 s coated magnet. In Figure 9 (a1,a2,b1,b2), the coating surface with Mg^2+^ added showed regular gully morphology with distributed micro-cracks compared with the coating without a charging agent, which is profit for the release of stress [24]. As is seen in Figure 9 (c1,c2), the coating surface with Cu^2+^ added exists aggregates on the surface, and shows finer gully morphology, which may be the reason that the coating adhesion was between without charging agents and with Mg^2+^ charging agents.

### 3.3. Magnetic Performance

The addition of Mg^2+^ and Cu^2+^ can improve the EPD efficiency and the adhesion of the coating, but it cannot be at the expense of deteriorating the magnetic properties after heat treatment. A high-temperature permanent magnet measuring instrument (NIM-500C) was used to measure the magnetic properties of the permanent magnetic materials at room temperature and high temperatures, respectively. An external field of ~3 T was initially applied to make the samples magnetized at full saturation. The remanence (B_r_), coercivity (H_cj_) and maximum magnetic energy product (BH)_max_ of the magnet were obtained by testing the demagnetization curve of the material. Figure 10 shows the demagnetization curves of the magnet, where it can be seen that the coercivity (H_cj_) of the original magnet was 19.3 kOe. With the first heat treatment process of 900 °C for 7 h, and the second heat treatment process of 500 °C for 2 h, the coercivity of the undeposited reference sample decreased to 18.1 kOe, while the coercivity of the EPD sample and EPD sample with Mg^2+^ and Cu^2+^ increased to 22.3 kOe, 22.2 kOe and 22.6 kOe, respectively. The remanence of all samples did not decrease. It indicates that the addition of Mg^2+^ and Cu^2+^ had no effect on the TbF_3_ coating GBD process. As an aside, the preparation of TbF_3_ coating containing copper has some development prospects, because Cu can promote GBD by widening the grain boundary [27]. 

## 4. Conclusions

Nano-TbF_3_ coating on the surface of the magnet was prepared with the addition of Mg^2+^ and Cu^2+^ in the EPD suspension, and the effects of Mg^2+^ and Cu^2+^ on EPD efficiency and coating adhesion were studied. The conclusions were as follows:(1)The addition of Mg^2+^ and Cu^2+^ can improve the EPD efficiency. With 3% relative content of Mg^2+^ added, the efficiency improved 116% from 3.1 mg/(cm^2^/min) to 6.7 mg/(cm^2^/min). With 5% relative content of Cu^2+^ added, the efficiency improved 109% from 3.1 mg/(cm^2^/min) to 6.47 mg/(cm^2^/min).(2)The effect of Mg^2+^ and Cu^2+^ on the EPD efficiency from the perspective of kinetics was analyzed with the Hamaker equation and diffusion double layer theory; it was found that Mg^2+^ and Cu^2+^ influence the EPD efficiency by changing the zeta potential of charged particles, but not the viscosity of suspension. In addition, the diffusion electric double layer absorbs Mg^2+^ or Cu^2+^ to increase its thickness, which indicates higher zeta potential of TbF_3_ particles when the relative content of Mg^2+^ and Cu^2+^ is low. When the relative content reaches 3% and 5%, respectively, the diffusion electric double layer reaches saturation, and further addition of a charging agent will compress the diffusion electric double layer and reduce zeta potential. Furthermore, the reduction reaction of Cu^2+^ is the reason of the lower rate of later EPD when Cu^2+^ charging agents were added.(3)It was found that the addition of Mg^2+^ and Cu^2+^ charging agents greatly improve the coating adhesion; the critical load for the cracking of the coating increased to 146.4 mN and 40.2 mN from 17.9 mN, respectively. Furthermore, the addition of Mg^2+^ and Cu^2+^ has no bad effect on TbF_3_ coating GBD of the magnet.

## Figures and Tables

**Figure 1 materials-16-02682-f001:**
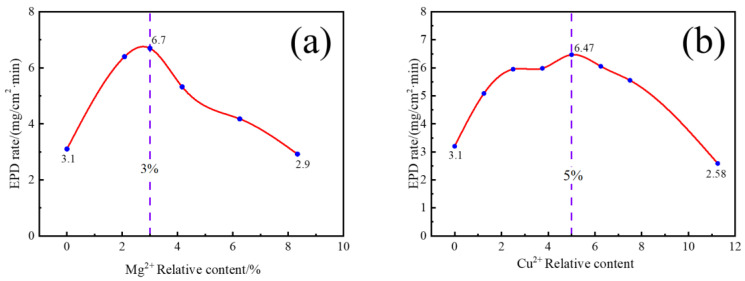
EPD rate of nano-TbF_3_ with different relative content of Mg^2+^ (**a**) and Cu^2+^ (**b**).

**Figure 2 materials-16-02682-f002:**
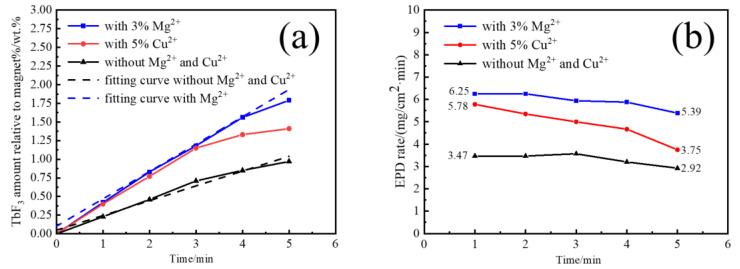
EPD TbF_3_ amount (**a**) and EPD rate (**b**) of nano-TbF_3_ at different EPD times.

**Figure 3 materials-16-02682-f003:**
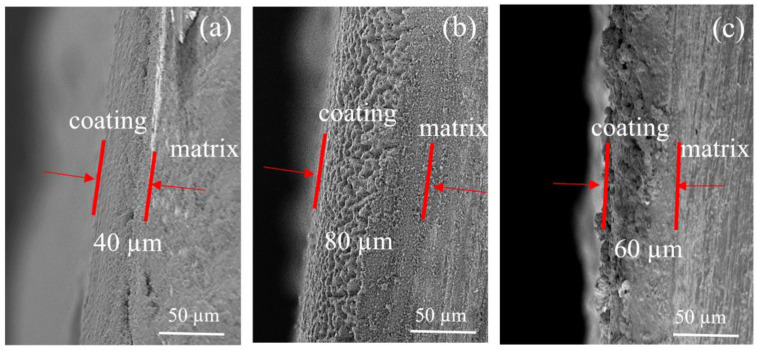
Cross-sectional SEM morphologies of the EPD 90 s coated magnet. (**a**) Without Mg^2+^ and Cu^2+^; (**b**) with Mg^2+^; (**c**) with Cu^2+^.

**Figure 4 materials-16-02682-f004:**
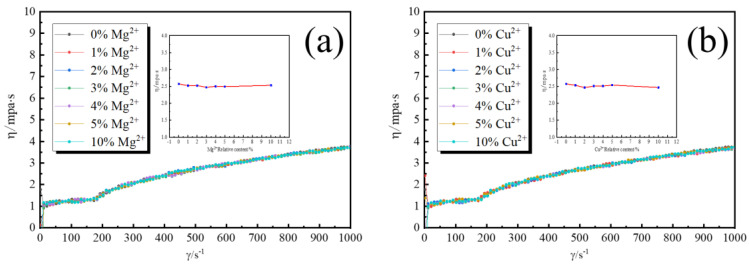
Viscosity of the EPD suspension with different Mg^2+^ and Cu^2+^ relative content changes with shear rate (inset is the average viscosity with different Mg^2+^ and Cu^2+^ relative content). (**a**) with Mg^2+^ (**b**) with Cu^2+^.

**Figure 5 materials-16-02682-f005:**
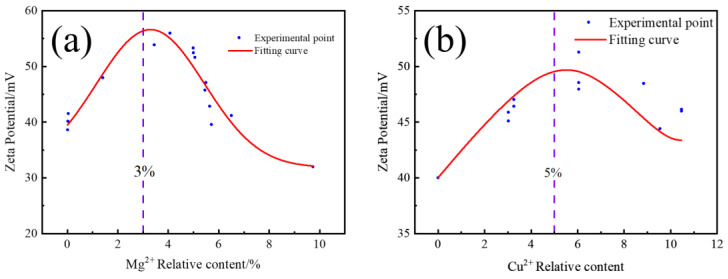
Zeta potential of EPD suspension with different relative contents of Mg^2+^ (**a**) and Cu^2+^ (**b**).

**Figure 6 materials-16-02682-f006:**
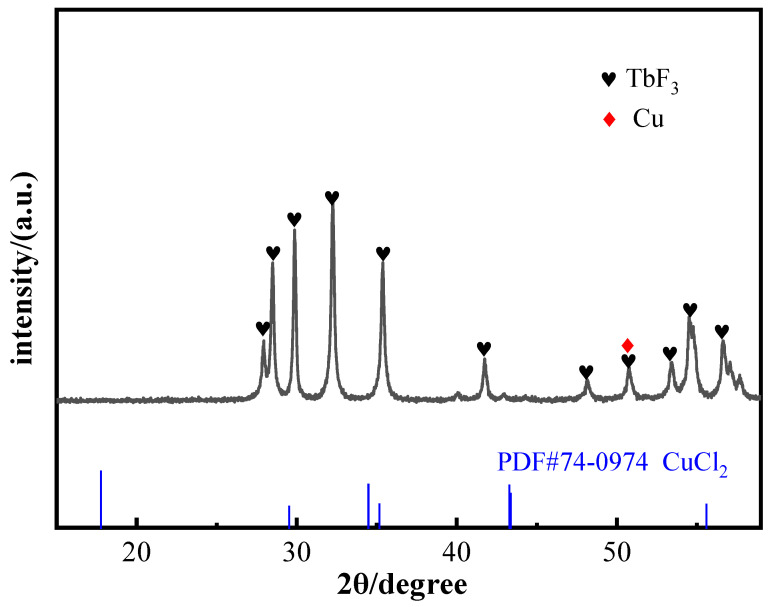
XRD results of EPD coating with Cu^2+^ charging agents.

**Figure 7 materials-16-02682-f007:**
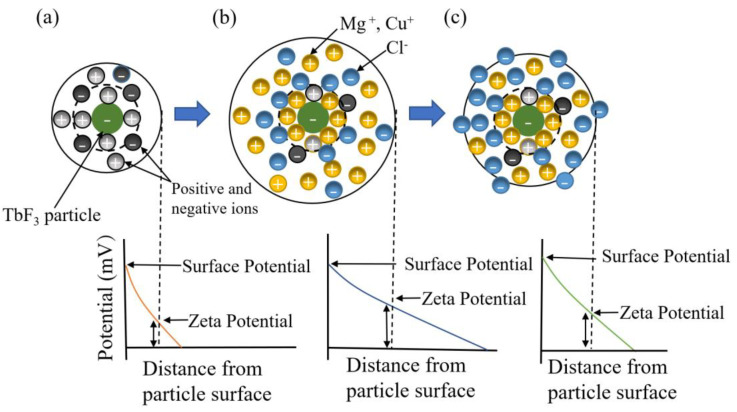
Schematic of the diffusion electric double layer region of a TbF_3_ particle. (**a**) Without charging agents; (**b**) with low content of charging agents; (**c**) with excess content of charging agents.

**Figure 8 materials-16-02682-f008:**
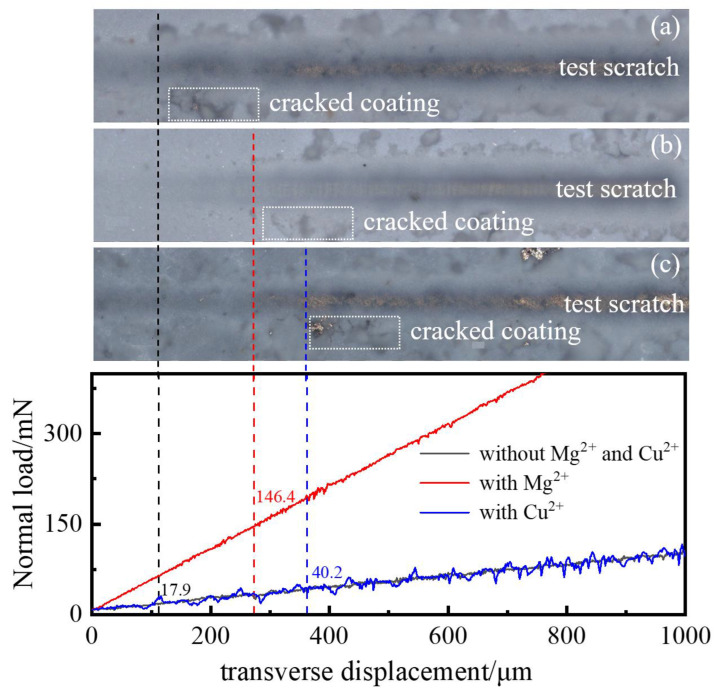
Scratch test results of EPD coating. (**a**) Without Mg^2+^ and Cu^2+^; (**b**) with Mg^2+^; (**c**) with Cu^2+^.

**Figure 9 materials-16-02682-f009:**
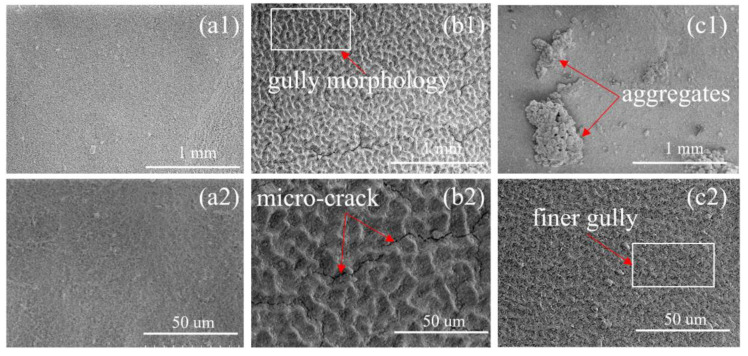
Surface SEM morphologies of the EPD 90 s coated magnet. (**a1**,**a2**) Without Mg^2+^ and Cu^2+^; (**b1**,**b2**) with Mg^2+^; (**c1**,**c2**) with Cu^2+^.

**Figure 10 materials-16-02682-f010:**
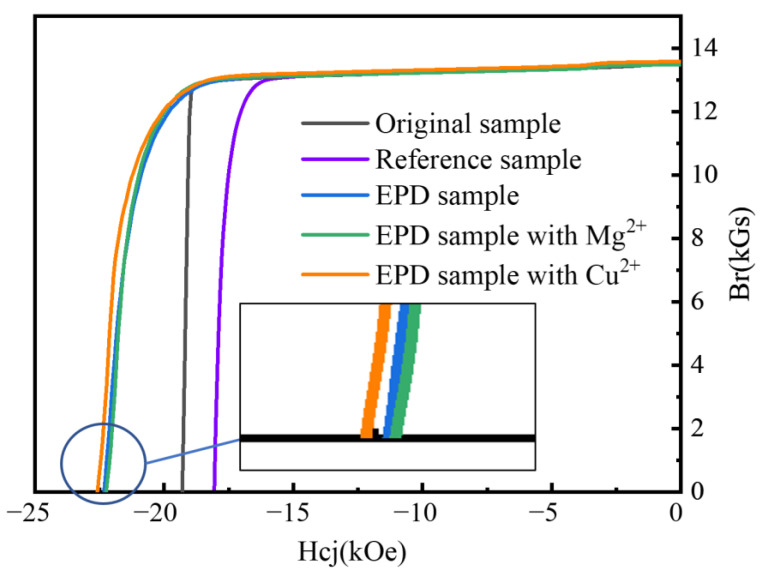
Demagnetization curves of magnet.

**Table 1 materials-16-02682-t001:** ICP element analysis results of coating.

Charging Agents	Tb	Cu
Mg^2+^	66.49	0.00
Cu^2+^	68.65	0.71

## Data Availability

Not applicable.

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
