# Peer review of "Mg2+ and Cu2+ Charging Agents Improving Electrophoretic Deposition Efficiency and Coating Adhesion of Nano-TbF3 on Sintered Nd-Fe-B Magnets"

_materials, 2023, doi:10.3390/ma16072682_

Round 1

Reviewer 1 Report

The authors present an exhaustive work on magnetic coatings based on Heavy metals.

Following are my observations:

1. Was the magnet surface observed under optical microscope or any other HD microscope to validate its smoothness/ morphology after sandpaper polish? If yes, request the authors to add a scan image and comment on the surface roughness.

2. A thorough correlation between EPD efficiency and Mg 2+, Ca 2+ ions is not well established throughout the results and discussion part. A qualitative approach is required to prove the author's claim. What is the window of Mg2+/Ca2+ % that can result in coercivity as well as an enhanced EPD?

3. Is there a correlation among all Group 2 elements on the EPD efficiency? 

Reviewer 2 Report

In the abstract as well as in the introduction section, the authors should enhance the clarity on what was the main objective of the work. The phrases shall be rephrased to give the reader a clear understanding of the goals.

In the experimental section, the authors mention about cutting the magnets into a cuboid of a size. What was the magnetization condition during the cutting process. The authors should mention the tooling used for the process. Can the authors provide more clarity in the regard?

Why did the authors test the viscosity up to a shear rate of 1000? Can the authors infer on the behaviour of the suspension based on the rheological data? Can the authors try to derive the log plots of the viscosity shear rate data for enhanced understanding and visualization?

What was the sample type for XRD analysis? Was the test on coatings done separately or was it on the magnet?

Can the authors label features observed in the SEM images?

Other images are significantly clear and are well labelled.

Can the authors elaborate more on the magnetic characterization? What type of a characterization equipment is that? Additionally field applied in testing should be elaborated. 

Round 2

Reviewer 1 Report

The manuscript maybe accepted in its present form.